# Direct Measurement of the Four-Phase Equilibrium Coexistence Vapor–Aqueous Solution–Ice–Gas Hydrate in Water–Carbon Dioxide System

**DOI:** 10.3390/ijms24119321

**Published:** 2023-05-26

**Authors:** Anton Semenov, Rais Mendgaziev, Andrey Stoporev, Vladimir Istomin, Timur Tulegenov, Murtazali Yarakhmedov, Andrei Novikov, Vladimir Vinokurov

**Affiliations:** 1Department of Physical and Colloid Chemistry, Gubkin University, 65, Leninsky Prospekt, Building 1, 119991 Moscow, Russia; meda810@mail.ru (R.M.); vlistomin@yandex.ru (V.I.); tulegenov08@list.ru (T.T.); murtazali99@bk.ru (M.Y.); novikov.a@gubkin.ru (A.N.); vladimir@vinokurov.me (V.V.); 2Department of Petroleum Engineering, Kazan Federal University, Kremlevskaya Str. 18, 420008 Kazan, Russia; 3Skolkovo Institute of Science and Technology (Skoltech), Nobelya Str. 3, 121205 Moscow, Russia

**Keywords:** gas hydrates, carbon dioxide, phase equilibria, ice, quadruple point, measurement

## Abstract

Precise data on the non-variant equilibrium of the four phases (vapor–aqueous solution–ice–gas hydrate) in *P*–*T* coordinates are highly desired for developing accurate thermodynamic models and can be used as reference points (similar to the triple point of water). Using the two-component hydrate-forming system CO_2_–H_2_O, we have proposed and validated a new express procedure for determining the temperature and pressure of the lower quadruple point Q_1_. The essence of the method is the direct measurement of these parameters after the successive formation of the gas hydrate and ice phases in the initial two-phase gas–water solution system under intense agitation of the fluids. After relaxation, the system occurs in the same equilibrium state (*T* = 271.60 K, *P* = 1.044 MPa), regardless of the initial parameters and the order of crystallization of the CO_2_ hydrate and ice phases. Considering the combined standard uncertainties (±0.023 K, ±0.021 MPa), the determined *P* and *T* values agree with the results of other authors obtained by a more sophisticated indirect method. Validating the developed approach for systems with other hydrate-forming gases is of great interest.

## 1. Introduction

The interaction between water and low molecular weight substances (carbon dioxide, lower hydrocarbons, nitrogen, hydrogen sulfide, etc.) at low temperatures/high pressure leads to the formation of crystalline inclusion compounds related to clathrate hydrates. In these compounds, the hydrate-forming molecules (guest) occupy the cavities of the crystal lattice built up by water molecules (host) [1]. The physicochemical properties of clathrate hydrates and their applications are discussed in recent reviews [2,3]. Recently, the study of the phase equilibria of carbon dioxide hydrate has received special attention due to its significant fundamental and applied relevance. From an applied standpoint, the formation of carbon dioxide hydrates is promising for seawater desalination [4,5,6], wastewater treatment [7], the prevention of CO_2_ emissions to the atmosphere through its sequestration and capture [8,9,10,11], the separation of gas mixtures [12,13,14,15], and in food industry technologies [16]. Carbon dioxide hydrate is a promising phase change material for cold energy storage [17,18]. A deeper understanding of the phase behavior in the CO_2_–H_2_O system is crucial for explaining processes in the lithosphere of Mars and other astronomical objects [19,20,21].

Experimental data on the phase behavior of carbon dioxide hydrates are necessary to predict the stability of the phases under certain conditions, which are the basis for the development of hydrate-based technologies. Data on different types of phase equilibria in the CO_2_–H_2_O system can be found in the literature. We will use the following symbols to denote the equilibrium phases: V—carbon dioxide-rich vapor phase; L_w_–water-rich liquid phase; L_w_*—supercooled water-rich liquid phase; L_CO2_—carbon dioxide-rich liquid phase; I—hexagonal ice; H—carbon dioxide hydrate. Based on the Gibbs rule of phases in the case of two independent variables (temperature and pressure), the number of degrees of freedom for any three-phase equilibrium in a two-component system is one. Such an equilibrium must correspond to a line in the phase diagram in the *P*–*T* coordinates. The V-L_w_-H monovariant three-phase equilibrium is the most common and has the greatest practical importance. The V-L_w_-H equilibrium temperatures and pressures for the CO_2_–H_2_O system have been experimentally determined by Ng and Robinson [22], Adisasmito et al. [23], Dholabhai et al. [24], Fan and Guo [25], Yang et al. [26], Wendland et al. [27], Mohammadi et al. [28], Yasuda and Ohmura [29], Melnikov et al. [30], Sami et al. [31], Nema et al. [32], Adeniyi et al. [33], Tariq et al. [34], Semenov et al. [35,36], and Cruz-Castro et al. [37]. Sun et al. [38] investigated the effect of a magnetic field on the V-L_w_-H equilibrium for carbon dioxide hydrate. The authors showed that the impact of the magnetic field with an induction of 0.39 T leads to a shift of the hydrate phase boundary by up to 3 K to a higher temperature region. Affecting the equilibrium conditions of hydrate formation with a magnetic field may develop a new method for accelerating hydrate formation and stabilizing the hydrate phase. However, the cited results need to be verified by other authors. Anderson [39] calculated the enthalpy of dissociation Δ*H*_d_ of carbon dioxide hydrate into water and gas, and the hydrate number *n* using the Clapeyron equation from the V-L_w_-H equilibrium data. Over the whole range of the V-L_w_-H phase coexistence, the Δ*H*_d_ value ranged from 63.6 ± 1.8 to 57.7 ± 1.8 kJ/mol, and the hydrate number varied from 6.6 ± 0.3 to 5.6 ± 0.3.

At temperatures below the H_2_O melting point, ice is a more thermodynamically stable phase than water, so the main type of equilibrium for such conditions is a monovariant V-I-H. For the considered CO_2_–H_2_O binary system, the temperatures and pressure of the coexistence of gas, ice, and hydrates have been measured by Miller and Smythe [40], Adamson and Jones [41], Wendland et al. [27], Yasuda and Ohmura [29], Fray et al. [42], Mohammadi and Richon [43], and Nagashima et al. [44]. V-I-H and V-L_w_-H equilibria for carbon dioxide hydrates with different isotopic compositions of ^12^CO_2_ and ^13^CO_2_ were studied by Kimura et al. [45]. It was found that the difference in the position of the equilibrium curves for ^12^CO_2_ and ^13^CO_2_ hydrates is small and is 0.007–0.012 MPa in the range of 269–278 K. At the same time, the ^12^CO_2_ hydrate has a slightly lower equilibrium pressure than ^13^CO_2_. Takeya et al. applied direct space methods combined with the Rietveld analysis to study the crystal structure of host–guest materials, including carbon dioxide hydrates [46]. The relevant studies of the phase equilibria of gas hydrates at temperatures below the freezing point of water are discussed in reviews [47,48].

At temperatures below the melting point of H_2_O, a metastable three-phase equilibrium of V-L_w_*-H with water supercooled relative to the ice phase is possible. This equilibrium is a metastable continuation of the V-L_w_-H line into the lower temperature region. Experimental data on the V-L_w_*-H equilibrium for the CO_2_–H_2_O system have been obtained by Wendland et al. [27], Melnikov et al. [30], and Nema et al. [32]. The dissociation of hydrates at temperatures < 273 K can lead to the formation of an ice layer on the surface of the hydrate particle, dramatically reducing the rate of the subsequent decomposition of the hydrate (the phenomenon of self-preservation of gas hydrates). Several studies have shown that gas hydrates initially decompose into gas and supercooled water, which can then turn into ice. The formation and long-term existence of supercooled water and gas hydrates in a metastable state has been documented by visual observations of hydrate dissociation at temperatures below the H_2_O melting point not only for carbon dioxide [30] but also for other gases such as propane, methane [49,50,51], and ethane [52]. The formation of supercooled water during hydrate dissociation has been confirmed using Raman spectroscopy for the ethane hydrate [52] and nuclear magnetic resonance relaxation spectroscopy for the difluorodichloromethane hydrate [53].

The ice melting curve in a carbon dioxide atmosphere (V-L_w_-I equilibrium) has been studied by Melnikov et al. [54]. The authors found that for the CO_2_–H_2_O system, the three-phase V-L_w_-I equilibrium line has a slope d*T*/d*P* that is 20 times greater (in modulo) than the L_w_-I equilibrium line of the one-component system with pure water. This is due to the high solubility of carbon dioxide in water, which lowers the equilibrium temperature of ice melting compared to the system without CO_2_. Melnikov et al. [54] also observed a metastable three-phase equilibrium V-L_w_*-I (at a pressure > 1.04 MPa) when the liquid water-rich phase is supercooled relative to the carbon dioxide hydrate phase. Although, according to the Schreinemaker’s method [55], an L_w_-H-I equilibrium should also be observed in this region, to the best of our knowledge, there is no such experimental data. We will be grateful to the reviewers and readers if they provide such data.

The peculiarity of carbon dioxide as a hydrate-forming gas is that at temperatures and pressures typical of the hydrate formation, CO_2_ is not a supercritical fluid and, therefore, can exist in either gas or liquid form. This complicates the phase behavior of the CO_2_–H_2_O system compared to similar systems containing non-condensing gases under the conditions studied. Reference data on the vapor–liquid equilibrium of pure carbon dioxide over a wide range of temperatures and pressures have been obtained by Duschek et al. [56]. Experimental data on V-L_CO2_-H and V-L_CO2_-L_w_ three-phase equilibria for the CO_2_–H_2_O system have been reported by Larson [57], Vlahakis et al. [58], Fan and Guo [25], and Wendland et al. [27]. Our comparative analysis showed that the experimental points of the two-phase equilibrium of the one-component system with CO_2_ [56] coincide with the data for the three-phase equilibrium V-L_CO2_-H and V-L_CO2_-L_w_ in the two-component CO_2_–H_2_O system [57,58] within the measurement uncertainties. At pressures above the V-L_CO2_-H and V-L_CO2_-L_w_ lines, the thermodynamic stability of carbon dioxide hydrate is determined by the L_CO2_-L_w_-H three-phase equilibrium. Experimental data for the latter were obtained by Takenouchi and Kennedy [59], Ng and Robinson [22], Fan and Guo [25], Yang et al. [26], and Adeniyi et al. [33]. In [60], the authors proposed a thermodynamic model to describe the phase behavior of gas hydrates in the system H_2_O and CO_2_ in a wide range of temperatures 150–295 K and pressures up to 500 MPa.

Figure 1 shows a fragment of the phase diagram for the two-component CO_2_–H_2_O system, constructed according to the above literature data, for the pressure and temperature range of 0.08–8 MPa and 220–300 K (with a logarithmic pressure scale). Appendix A shows the same phase diagram with a linear pressure scale. At the lower quadruple point Q_1_, the lines of the monovariant equilibria V-L_w_-H, V-I-H, and V-L_w_-I intersect. Four phases V-L_w_-I-H coexist at this point, so this equilibrium is non-variant because the number of degrees of freedom must equal zero according to the Gibbs phase rule for a two-component system. Therefore, the four phases V-L_w_-I-H can only be in equilibrium at the fixed pressure and temperature values. Precise temperature and pressure data at the quadruple point of the CO_2_–H_2_O system are valuable for developing accurate thermodynamic models. They can also be used as a reference point (analogous to the triple point of water).

The Q_1_ point coordinates are usually determined by experimentally measuring two types of equilibria, V-L_w_-H and V-I-H [29], V-L_w_(L_w_*)-H and V-I-H [32], or V-L_w_(L_w_*)-I and V-I(L_w_)-H [54], approximating the experimental points for each equilibrium by a function (usually, ln*P* = *A* + *B*·*T*^−1^), and finding the intersection of the two functions. This procedure requires a considerable amount of time and other resources.

In this paper, we report the results of our experimental observations, which indicate the possibility of a well-reproducible measurement of pressure and temperature at the lower quadruple point Q_1_ in a single experiment, using the example of a CO_2_–H_2_O system with the successive crystallization of gas hydrates and ice (or ice and then gas hydrate) under intense stirring and good heat transfer in the system. Our results suggest reducing the time required to determine the equilibrium conditions of the four-phase equilibrium coexistence of the vapor–aqueous solution–ice–gas hydrate and significantly reducing the measurement uncertainty.

## 2. Results

To confirm the validity of the pressure and temperature readings of the GHA350 setup used for the CO_2_ hydrate equilibrium measurements, we examined two reference systems with a priori known equilibrium *P* and *T* parameters. As such comparison references, we chose the freezing point of water at atmospheric pressure and the equilibrium pressure/temperature of V-L_w_-H coexistence in the CO_2_–H_2_O system.

### 2.1. Measurement Results of Reference Systems

#### 2.1.1. Ice Freezing Point at Ambient Pressure

Figure 2 shows the temperature curve obtained by measuring the ice freezing point of distilled water with a Pt100 resistance thermometer from a GHA350 autoclave under intensive agitation at 600 rpm. The temperature in the system was lowered at a rate of 3 K/h (linear section starting at 30 min). Crystallization of the supercooled water started at 271.18 K, after which the temperature rapidly increased (10 s) to 273.14 K and remained at a plateau. The mean and standard deviation of the temperature at the ice freezing plateau were 273.144 ± 0.007 K. The average value differs from the reference ice melting temperature of 273.1525 K (at 101,325 Pa) [61] by no more than 0.0085 K. It is also clear from the data in Figure 1 that the measured temperature fluctuations are in the range of no more than 0.03 K (from 273.13 to 273.16 K).

#### 2.1.2. Three-Phase Equilibrium Conditions V-L_w_-H of CO_2_–H_2_O System

Figure 3a shows the experimental values of the equilibrium conditions for the coexistence of the gaseous CO_2_-rich phase, the aqueous solution, and the CO_2_ hydrate measured by us [35,36], as well as the literature data [23,24,28,29,30,31,32]. As seen, there is good agreement between our values and the literature.

To derive the correlation between the pressure and temperature at the V-L_w_-H equilibrium line for the H_2_O–CO_2_ system, we used Equations (1) [36] and (2):(1)$$P=\exp(A+\frac{B}{T}+C\cdot \ln T),$$
(2)$$P=A_{0}+A_{1}T+A_{2}T^{2}+A_{3}T^{3}+A_{4}T^{4},$$
where *P* and *T* are the equilibrium pressure and temperature, respectively; *A*, *B*, *C*, and *A*_0_–*A*_4_ are the fitted parameters. The approximation results are shown in Figure 3a as solid and dashed lines. The points in Figure 3b illustrate the difference between the experimental and calculated equilibrium pressure as a function of temperature for each of the two models. The numerical values of the coefficients are given in Appendix A. As expected, the polynomial Equation (2) is more flexible compared to model 1, and the residuals plot (Figure 3b) shows that the difference between the experimental and calculated values is several times smaller in the case of Equation (2), especially at higher pressures. From the data in Appendix A, we can see that Equation (1) has an average absolute deviation (AAD) of 0.016 MPa (0.63%), while for Equation (2), this value is 0.004 MPa (0.19%). Thus, the accuracy of the description of the experimental data increases by a factor of 3.5 in the transition from model 1 to 2.

To quantify the differences between the results of our measurements of the V-L_w_-H equilibrium conditions for the carbon dioxide hydrate and the literature data, we calculated the relative difference using Equation (3):(3)$$\delta =\frac{P_{\exp}-P_{\mathrm{fit}}}{P_{\mathrm{fit}}}\times 100\%,$$
where *P*_exp_ is the experimental value of the equilibrium pressure, *P*_fit_ is the calculated value of the equilibrium pressure obtained by substituting the experimental equilibrium temperature *T*_exp_ into approximations 1 and 2. Both approximations were derived only from our experimental data [35,36]. The calculated relative differences of equilibrium pressure from Equation 3 are shown in Figure 4 and Figure 5.

The comparison in Figure 4 and Figure 5 shows that our measured equilibrium pressures for the carbon dioxide hydrate [35,36] are in best agreement with the results of Adisasmito et al. [23] and Dholabhai et al. [24]. The equilibrium pressures of Adisasmito et al. [23] are overestimated by 0.49% on average compared to our values [35,36], and the points of Dholabhai et al. [24] are overestimated by 0.85%. The experimental points of Nema et al. [32] (*δ* = 1.21%) are also shifted to slightly higher pressures. The experimental points of Mohammadi et al. [28], Yasuda and Ohmura [29], Melnikov et al. [30], and Sami et al. [31] (*δ* = −1.17%, −1.53%, −1.58%, and −1.59%, respectively) are underestimates of the equilibrium pressure relative to our data.

Summarizing the results obtained for the second reference system (V-L_w_-H equilibrium in the CO_2_–H_2_O system), we can conclude a good agreement between our measured equilibrium temperatures and pressures [35,36] and the literature data [23,24,28,32]. There is also a satisfactory agreement of our results with the data [29,30,31].

### 2.2. Measurement of Pressure and Temperature in the Lower Quadruple Point Q_1_ of CO_2_–H_2_O System

Figure 6 shows the experimental curves of the pressure and temperature versus the time obtained in the study of the hydrate equilibria in the CO_2_–H_2_O system [35,36]. The curves are not shown for the entire time range but only for the hydrate formation stage during cooling. The experiments were carried out using a GHA350 autoclave equipped with a stirring system that allows high-intensity stirring of the fluids. The results of our analyses [36,62] show that for distilled water (without accounting for changes in density and viscosity due to gas dissolution) at a stirrer speed of 600 rpm and a temperature of 273–283 K, the Reynolds number is more than 20,000, i.e., the character of the flow in the GHA350 autoclave during operation of the stirring system is fully turbulent [63]. High-intensity fluid agitation eliminates the diffusion limitations of the hydrate formation reaction and contributes to the rapid heat transfer of the hydrate and ice crystallization. We performed six runs, all of which observed the sequential formation of carbon dioxide hydrate and ice (or vice versa, ice and then hydrate). Runs 1–6 differed in their initial conditions (*P*, *T*). Consider the curves of run 1 in more detail.

The initial pressure and temperature of the fluids in the autoclave in run 1 were 1.15 MPa and 273.26 K, corresponding to conditions beyond the CO_2_ hydrate stability zone. Hydrate formation was induced by cooling the system at a 5 K/h rate. CO_2_ hydrate growth started at 42.3 min (time marked with a black dashed line) and was followed by a short temperature peak (of 0.13 K) due to the release of heat from the carbon dioxide hydrate formation reaction and a concurrent kink in the pressure curve due to the onset of gas uptake. Hydrate growth then proceeded at a decreasing temperature (due to the difference between the temperature of the fluids in the autoclave and the temperature of the coolant circulating in the autoclave jacket). The temperature in the autoclave declined until ice crystallization occurred, which started at 63.4 min (time marked with a cyan dashed line). Due to ice freezing, the autoclave temperature increased by 1.55 K to 271.61 K in ≈ 1 min and remained at the plateau for over 1 h. The onset of ice crystallization also increased the pressure, which is associated with a reduction in the free volume of the system [64]. However, the pressure relaxation in the system was slower than the temperature relaxation. The pressure plateau after the onset of ice crystallization took about 6 min, which is six times longer than the time for the temperature to reach the plateau. In the range of 72.5–140.9 min with the simultaneous presence of four phases in the system (vapor, water solution, hydrate, and ice), it occurs while intensive stirring, pressure, and temperature are constant (271.60 ± 0.01 K, 1.044 ± 0.002 MPa), which can be interpreted as an equilibrium state. The temperature and pressure values at the plateau agree very well with the literature data on the coordinates of the lower quadruple point in the CO_2_–H_2_O system [29,32,54], which were determined indirectly. After 140.9 min, a trend of slow increases in the pressure and decreases in the temperature is observed, which we attribute to the accumulation of carbon dioxide hydrates and ice in the system. At a certain amount of solids, the stirrer can no longer effectively mix the fluids in the system, resulting in a gradual shift of the system state away from equilibrium for the four phases V-L_w_-I-H.

Runs 2–5 (Figure 6b–e) differ from run 1 in the higher initial pressure in the system (more CO_2_ in the autoclave). However, the cooling curves for runs 2–5 are qualitatively the same as run 1. The carbon dioxide hydrate begins to crystallize first during cooling, followed by ice. During the transition from run 1 to run 5, the regularity of the increase in time between the onset of hydrate and ice crystallization (21.1 min for run 1 and 87.6 min for run 5) is evident. Ice freezing starts at approximately the same temperature (269.8–271.3 K) in the considered experiments. Since both the initial *T* and *P* increase from run 1 to run 5, an expected consequence is an increase in the cooling time of the ice nucleation temperature. In the case of runs 2–5, as a result of the successive appearance of hydrate and ice in the system, the temperature and pressure at the plateau (after relaxation) take the same values within the measurement errors as in run 1, coinciding with the numerical values of parameters of the lower quadruple point of the CO_2_–H_2_O system [29,32,54].

For run 6, the initial parameters were close to those of run 1, but the shape of the pressure and temperature curves was qualitatively different from runs 1–5. The difference for run 6 was that the ice phase started to crystallize at 100.6 min, resulting in a rapid temperature rise of 2.6 K (from 269 K to 271.6 K) for 2 min. The carbon dioxide hydrate began to form 3 min after the appearance of ice. The occurrence and growth of ice and then CO_2_ hydrate caused the pressure to increase to a peak value of 1.159 MPa and the subsequent relaxation of *P* to 1.045 MPa. The second thermal effect on the temperature curve caused by the appearance of the CO_2_ hydrate phase was absent in the case of run 6. From the curve for run 1, the exothermic effect of hydrate crystallization is an order of magnitude smaller in amplitude than that of ice formation, in agreement with our previous results [65]. Therefore, the superposition of the two exotherms in run 6 resulted in the less intense temperature signal of hydrate formation not being detected. It took 14 min for the pressure to relax from a peak of 1.159 MPa to 1.045 MPa, after which the parameters remained constant (271.59 ± 0.01 K, 1.045 ± 0.002) for 82 min with four phases in the system: gaseous CO_2_, aqueous solution, ice, and carbon dioxide hydrate.

Thus, regardless of the initial parameters (*T* and *P*) and the order in which the CO_2_ hydrate and ice phases occur during cooling, the system appears in the same equilibrium state after relaxation with values of temperature 271.60 ± 0.01 K and pressure 1.044 ± 0.002, corresponding to the non-variant equilibrium of four phases V-L_w_-I-H in the CO_2_–H_2_O system.

The resulting *P*(*T*)-trajectories for runs 1–6 are shown in Figure 7a–f. Each panel of this figure also offers a point corresponding to the mean temperature and pressure value at the plateau according to the results of six independent measurements.

One can see from the data in Figure 7 that when varying the initial parameters *P* and *T* in quite a wide range, the experimental trajectory in each case crosses the equilibrium point of the four phases V-L_w_-I-H in the CO_2_–H_2_O system. This observation confirms the reproducibility and universal nature of the observed phenomenon.

## 3. Discussion

The results of the measurements of the equilibrium parameters at the quadruple point for the H_2_O–CO_2_ system are shown graphically in Figure 8; the numerical values of the pressure and temperature are in Table 1. One can conclude that the results from runs 1–6 have good repeatability.

The value of *T* and *P* for each run corresponds to the mean value of the parameter at the plateau (averaged over time). The average (last row in Table 1) is calculated by averaging the parameters over the six points obtained. Based on the results in Table 1, the metrological characteristics of the measuring instruments and standards used for calibration, we calculated the combined standard and expanded uncertainties of the equilibrium pressure and temperature according to JCGM 100:2008 [66] using Equations (4) and (5):(4)$$u_{c}=\sqrt{u_{1}^{2}+u_{2}^{2}+u_{3}^{2}},$$
(5)$$U=k\cdot u_{c},$$
where *u_c_* is the combined standard uncertainty of the equilibrium temperature or pressure, *u_1_* is the maximum error of the standards (0.011 K and 0.017 MPa) used to calibrate the pressure and temperature sensors of the GHA350 rig. A 1524 reference thermometer coupled with a PRT 5616-12 reference sensor (both Fluke, Everett, WA, USA) was utilized to calibrate the Pt100 temperature sensor. The pressure sensor was calibrated with a 717 5000G (Fluke, Everett, WA, USA). *u_2_* is the repeatability of the GHA350 sensors (0.019 K and 0.012 MPa). The repeatability of the Pt100 was evaluated by measuring the freezing point of water at atmospheric pressure three times (one measurement per day). The repeatability of the GHA350 pressure sensor was determined by comparing the difference in readings with the reference gauge 700G30 (Fluke, Everett, WA, USA) at 1.1–2.5 MPa (after completion of runs 1–6). *u_3_* is the standard deviation of the equilibrium pressure or temperature from runs 1–6 (0.006 K and 0.002 MPa; last row in Table 1). U is the expanded uncertainty of the equilibrium temperature or pressure, and *k* is the coverage factor, which was taken as two. The combined standard uncertainty of equilibrium temperature and pressure of Q_1_ point is 0.023 K and 0.021 MPa, and the expanded uncertainty (at 95% confidence level) is 0.045 K and 0.042 MPa.

A comparison of the results with the literature data is in Table 2. Taking into account the combined standard uncertainties (±0.023 K, ±0.021 MPa) in the temperature and pressure (see Figure 8b and Table 2), the measured parameters at the quadruple point agree with the literature data from Yasuda and Ohmura (2008) [29], Nema et al. (2017) [32], and Mel’nikov et al. (2014) [54]. It should be emphasized again that in this work, the temperature and pressure of the four-phase V-L_w_-I-H equilibrium were measured directly during the successive crystallization of hydrate and ice (or in reverse order) in the system. While in the cited papers [29,32,54], the coordinates of the Q_1_ point were determined indirectly by a more time-consuming method by measuring the equilibrium points of two different three-phase equilibria (see Table 2), approximating the obtained points by two equations, and determining their intersection point. Direct measurement of the four-phase equilibrium conditions reduced the combined standard uncertainty of the equilibrium temperature by an order of magnitude compared to literature data [29,32,54].

Let us discuss the possible reasons for directly measuring the *P* and *T* parameters of the four-phase equilibrium for the CO_2_–H_2_O system. First, the high turbulence of the GHA350 stirring system at 600 rpm (*Re* > 2 × 10^4^ [36,62,67]), which significantly intensifies the heat and mass transfer processes, contributes to the rapid reaching of the equilibrium state in the system. Second, in the simultaneous presence of the CO_2_ hydrate and ice phases in the system, the processes of their formation compete with each other [68]. When ice crystallizes in the system, the free volume decreases, increasing pressure [64]. Hydrate formation leads to the opposite effect of a pressure decrease due to the gas uptake and reduction in its amount in the free volume. Thus, the system’s equilibrium state (*P* and *T* at the plateau) with the simultaneous crystallization of hydrate and ice in the system can be explained by the rates of these processes that may be comparable. In this case, the system is self-regulating, i.e., the effect of the pressure increase due to ice crystallization is compensated by the pressure decrease due to gas hydrate formation. The ice freezing process usually occurs at a higher rate than the formation of the sI methane hydrate or sII C_1_-C_3_ hydrate [65]. However, CO_2_ has a higher solubility in water compared to hydrocarbons, so the kinetics of carbon dioxide hydrate formation are faster and more favorable [69,70,71,72]. Therefore, the difference between the ice and gas hydrate formation rates becomes smaller in the case of carbon dioxide.

Verification of the proposed technique on other similar systems will allow one to use it for the direct measurement of the conditions of the non-variant four-phase V-L_w_-I-H equilibrium of various hydrate-forming systems to obtain precise data for the construction and refinement of phase diagrams. In future work, we plan to investigate how the nature of the hydrate-forming gas and the type of hydrate structure affect the feasibility of directly measuring the four-phase equilibrium using the described technique. The nature of the gas and the type of hydrate structure (sI or sII) influence the kinetics of hydrate formation. Therefore, it is of interest to compare the CO_2_–H_2_O system (high gas solubility in water, sI hydrate) with others such as N_2_-H_2_O (lower gas solubility in water, sII hydrate) and CH_4_-H_2_O (lower gas solubility in water, sI hydrate). Special attention should also be paid to the study of the proposed technique for systems with kinetic promoters of hydrate formation [73,74], which allows the rate of hydrate formation to be significantly increased. This can reduce the difference in the rate of hydrate and ice growth processes. The direct measurement of the four-phase equilibrium in hydrate-forming systems according to the procedure described in the manuscript significantly improves the accuracy of the determination of the coordinates of the lower quadruple point. It reduces the time and material costs of research compared to conventional methods.

## 4. Materials and Methods

The pressurized carbon dioxide with at least 99.99 vol% of the main component was purchased from NIIKM (Moscow, Russia). The Simplicity UV laboratory water purification system produced deionized water with a resistivity of 18.2 MΩ·cm at 298.15 K (Merck Millipore, Burlington, MA, USA).

The studies were performed with a GHA350 apparatus (PSL Systemtechnik, Osterode am Harz, Germany) [75], which comprises a 600 mL Hastelloy autoclave rated at a peak operating pressure of 35 MPa. The autoclave is coupled to a Hei-TORQUE 400 precision drive (Heidolph, Schwabach, Germany), a Minipower magnetic coupling (Premex, Lyss, Switzerland), and a four-blade propeller (diameter of 6.1 cm). The temperature control system involves an outer jacket with an ethanol coolant whose circulation is provided by a Ministat 240 (Huber, Offenburg, Germany). Ministat 240 maintains coolant temperature stability within ±0.02 K. The autoclave is fitted with calibrated temperature transducer (Pt100) with a resolution of ±0.01 K.

A P3251 sensor (Tecsis, Offenbach am Main, Germany), in combination with a 700G30 electronic reference manometer (Fluke, Everett, WA, USA), is used to measure the pressure in the autoclave with a resolution of 0.001 MPa. The Pt100 resistance thermometer and the GHA350 pressure sensor were calibrated prior to the experiments using the reference instruments and the methodology described previously [76]. The sensors, the Hei-TORQUE 400 precision stirrer, and the Ministat 240 thermostat are connected to a PC running WinGHA software (version 4.0.10.790, PSL Systemtechnik, Osterode am Harz, Germany). After executing the WinGHA script, the setup operation is controlled, and the measured parameters are automatically collected. When the script is stopped, the data saved in the file with the .dat extension can be processed.

The autoclave, previously cleaned by washing with distilled water and blowing with compressed air, was filled with 300 mL of 18.2 MΩ·cm water. The empty volume was blown out three times with carbon dioxide to remove residual air. At 295 K, the autoclave was pressurized with gaseous CO_2_ and stirred at 600 rpm to accelerate the dissolution of the carbon dioxide in the water. Stirring was continued until the end of the experiment. The specified stirring speed was standard for us when measuring gas hydrate equilibria with the GHA350 [77]. After pressure stabilization, the autoclave temperature was lowered to a level where the *P* and *T* parameters were outside the hydrate stability zone but close to the V-L_w_-H curve for the CO_2_–H_2_O system [23,24,35,36]. The system was then held for 1 h to allow mutual saturation of the gas and liquid phases, after which the coolant temperature was reduced by 5–10 K at a rate of 5 K/h. After the ramp cooling, the thermostat maintained the coolant temperature at a constant level. Successive formation of CO_2_ hydrate and ice was detected by thermal effects and pressure changes.

## 5. Conclusions

We have proposed a new technique for the direct determination of the *P* and *T* coordinates of the lower quadruple point (non-variant four-phase equilibrium gas–water solution–ice–gas hydrate) on the example of the two-component system carbon dioxide–water. The technique directly measures the temperature and pressure after the successive appearance of gas hydrates and ice phases in the initial two-phase gas–water solution system under intensive fluid agitation. Irrespective of the initial parameters (*T* and *P*) and the order of hydrate and ice occurrence, the system is in the same equilibrium state after relaxation *T* = 271.60 ± 0.01 K (mean and standard deviation) and *P* = 1.044 ± 0.002 MPa. A good reproducibility of the measurement results from six independent experiments was demonstrated. The obtained values of the *P* and *T* parameters at the lower quadruple point for the CO_2_–H_2_O system, considering the combined standard uncertainty of the temperature and pressure measurements (±0.023 K, ±0.021 MPa), are in perfect agreement with the literature data [29,32,54] determined by a more complicated and indirect method. The direct measurement allows us to reduce the combined standard uncertainty of the equilibrium temperature by an order of magnitude compared to the literature data. It is of great interest to validate the developed technique for other hydrate-forming systems.

## Figures and Tables

**Figure 1 ijms-24-09321-f001:**
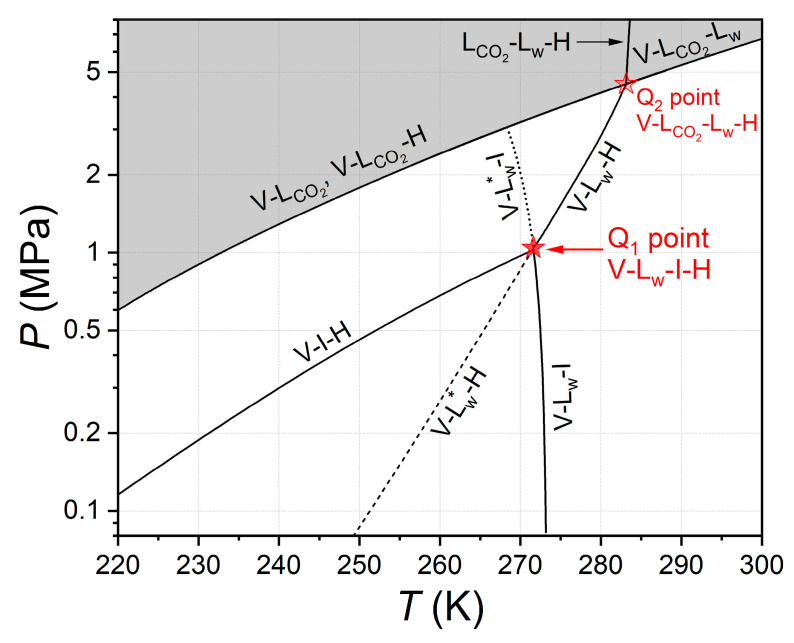
Phase diagram for the H_2_O–CO_2_ system based on literature data for the temperature and pressure range of 220–300 K and 0.08–8 MPa; gray and white colors correspond to the regions of existence of liquid and gaseous carbon dioxide, respectively; solid line V-L_w_-H is the three-phase gas–aqueous solution–gas hydrate equilibrium (fit based on [35,36]); dashed line V-L_w_*-H is the metastable three-phase gas–supercooled aqueous solution–gas hydrate equilibrium (fit based on [30,32]); solid line V-I-H is the three-phase gas–ice–gas hydrate equilibrium (fit based on [29,44]); solid line V-L_w_-I is the three-phase gas–liquid aqueous solution–ice equilibrium (fit based on [54]); dotted line V-L_w_*-I is the metastable three-phase gas–supercooled aqueous solution–ice equilibrium (fit based on [54]); solid line V-L_CO2_ is the two-phase equilibrium of gaseous and liquid carbon dioxide (fit based on [56]) overlapping with solid line V-L_CO2_-(H or L_w_) of the three-phase equilibrium of gaseous and liquid carbon dioxide and gas hydrate (or aqueous solution, fit based on [57,58]); solid line L_CO2_-L_w_-H is the three-phase liquid carbon dioxide-rich phase–aqueous solution–gas hydrate equilibrium (fit based on [33,59]); the red stars represent the non-variant four-phase equilibria: gas–aqueous solution–ice–gas hydrate (Q_1_ point (271.60 K, 1.044 MPa), data of this work) and gas–liquid carbon dioxide-rich phase–aqueous solution–gas hydrate (Q_2_ point, (283.13 K, 4.494 MPa), intersection of fits based on [35,36,57,58]).

**Figure 2 ijms-24-09321-f002:**
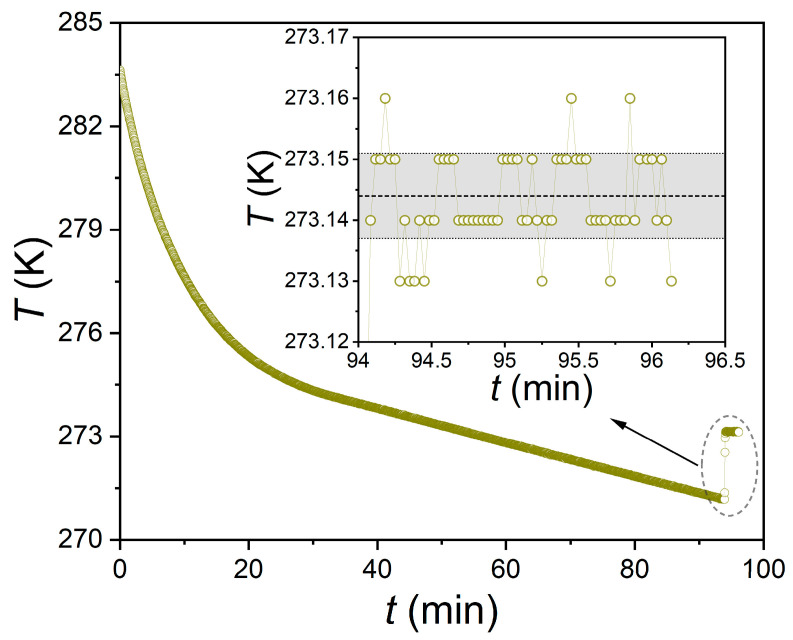
Temperature evolution in GHA350 autoclave during measurement of ice freezing point at ambient pressure.

**Figure 3 ijms-24-09321-f003:**
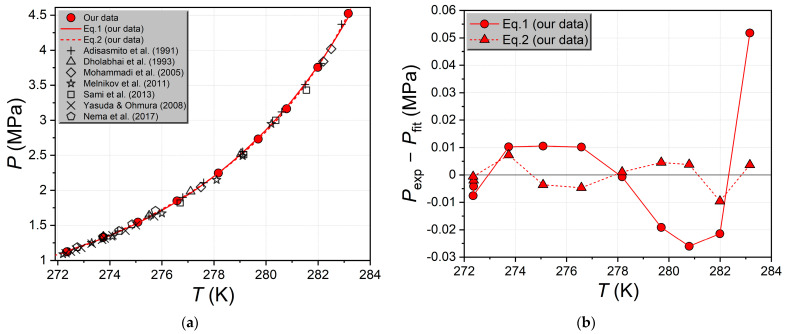
(**a**) Measured pressures and temperatures of the monovariant three-phase equilibrium V-L_w_-H of H_2_O–CO_2_ system; red circles—our data [35,36]; black symbols—literature data [23,24,28,29,30,31,32]; solid red line—fit from our data by three-parameter Equation (1) (see ref. [36]); red dashed line—fit from our data by a fourth-degree polynomial (see Equation (2)); (**b**) fitting residuals for both models as a function of equilibrium temperature.

**Figure 4 ijms-24-09321-f004:**
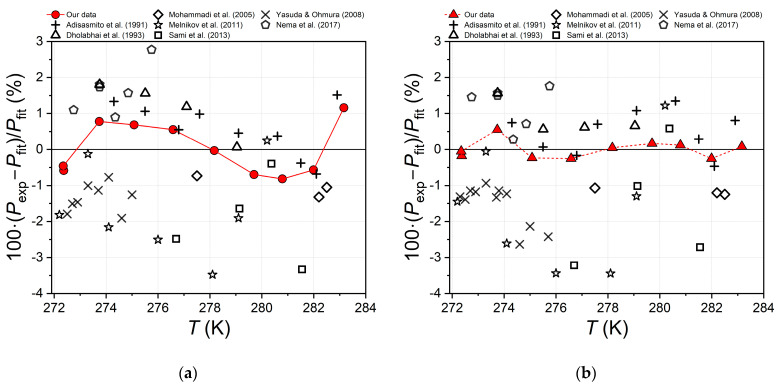
(**a**) The relative difference between the experimental [23,24,28,29,30,31,32,35,36] and calculated (approximation by Equation (1) using data from [35,36]) equilibrium pressure of carbon dioxide hydrate; (**b**) similar relative difference for the fit with Equation (2).

**Figure 5 ijms-24-09321-f005:**
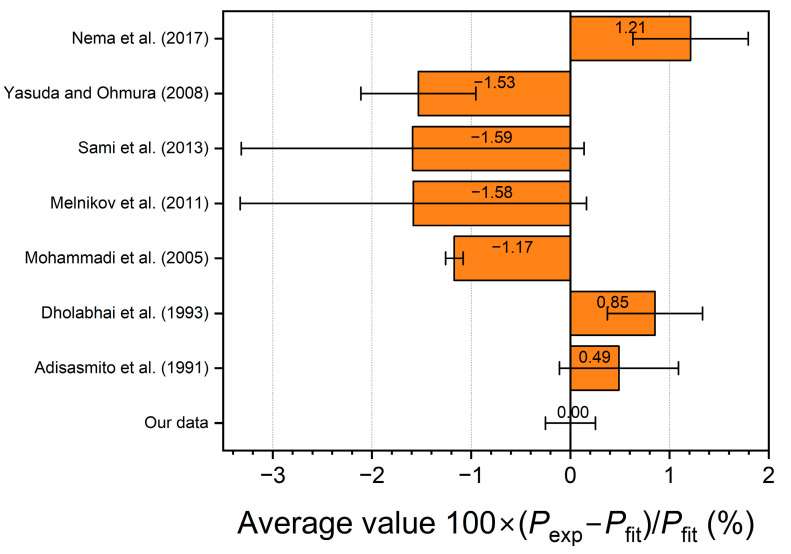
Average relative pressure difference in three-phase equilibrium V-L_w_-H in system CO_2_–H_2_O calculated by Equation (3) for polynomial approximation 2 (data in Figure 4b); experimental data are from [23,24,28,29,30,31,32,35,36]; error bars are standard deviations of average.

**Figure 6 ijms-24-09321-f006:**
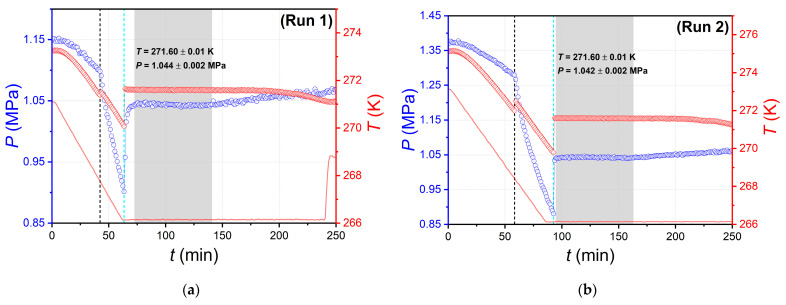
(**a**) Evolution of pressure and temperature over time in the GHA350 autoclave (blue and red symbols) in run 1; the onsets of CO_2_ hydrate formation and ice crystallization are shown by vertical black and cyan dashed lines, respectively; the solid red line reflects temperature change in the coolant circulating in the GHA350 autoclave jacket; the gray fill shows the time with steady temperature and pressure readings in the autoclave after the occurrence of hydrate and ice in the system and the next reaching a plateau of *T* and *P*; the variation of temperature and pressure at the plateau is displayed as the mean and standard deviation of *T* and *P*; (**b**–**f**) similar results from runs 2–6 at different initial pressure and temperature in the system.

**Figure 7 ijms-24-09321-f007:**
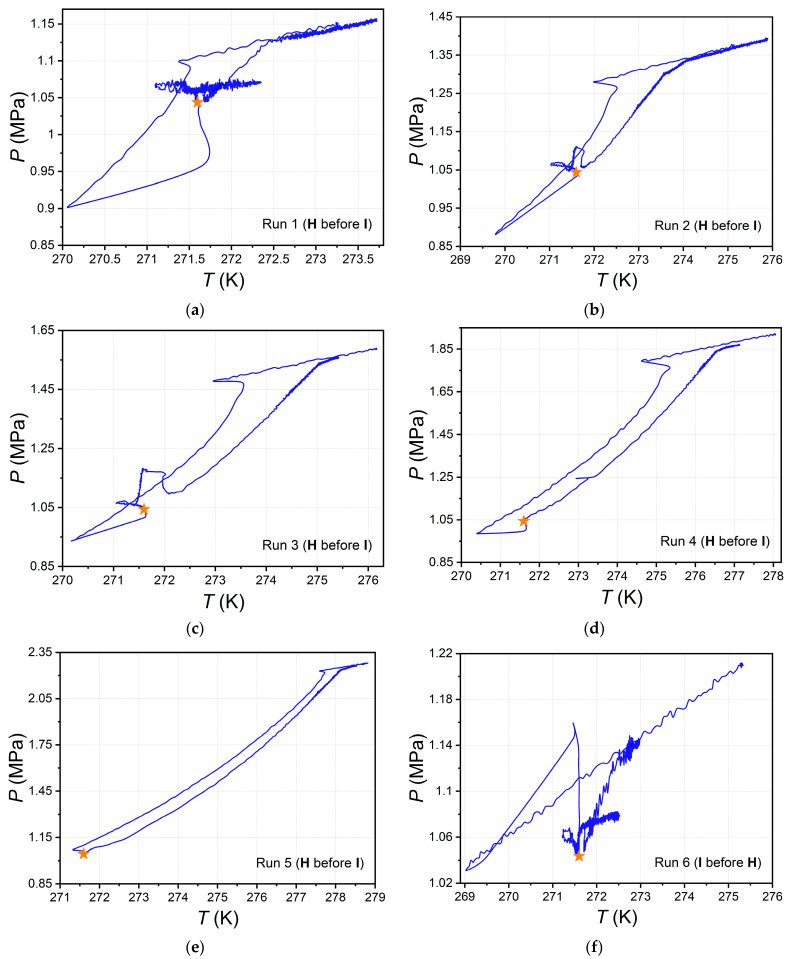
(**a**–**f**) Pressure–temperature trajectories from runs 1–6 (solid blue line); orange star depicts *P* and *T* at lower quadruple point (four phases in equilibrium V-L_w_-I-H) for H_2_O–CO_2_ system (see our data in Table 1).

**Figure 8 ijms-24-09321-f008:**
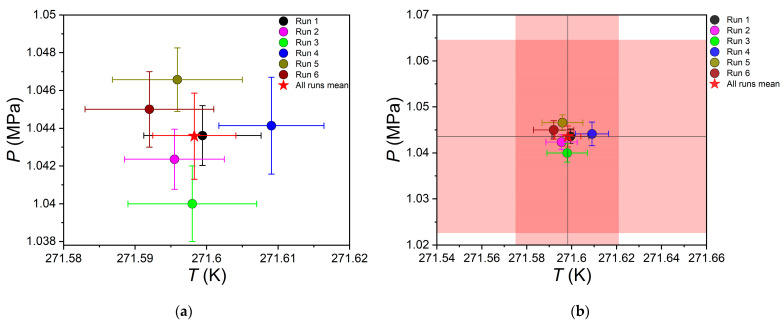
(**a**) Measured values of temperature and pressure at the plateau after the successive occurrence of hydrate and ice (runs 1–5) or ice and hydrate (run 6) in the H_2_O–CO_2_ system. Panel (**a**) also demonstrates the average temperature and pressure of the non-variant equilibrium (red star) calculated from data of all single measurements (runs 1–6); (**b**) similar quantities plotted considering the combined standard uncertainties in temperature (±0.023 K) and pressure (±0.021 MPa) for the average (runs 1–6). The uncertainties *u_c_* in temperature and pressure are shown as a light-red fill in the corresponding domain.

**Table 1 ijms-24-09321-t001:** The measurement results of runs 1–6 of the temperature and pressure of the four-phase equilibrium gaseous carbon dioxide–aqueous solution–ice–carbon dioxide hydrate.

Run	*T*, K	*P*, MPa	Standard Deviation *T*, K	Standard Deviation *P*, MPa
1	271.60	1.044	0.008	0.002
2	271.60	1.042	0.007	0.002
3	271.60	1.040	0.009	0.002
4	271.61	1.044	0.007	0.003
5	271.60	1.047	0.009	0.002
6	271.59	1.045	0.009	0.002
Average	271.60	1.044	0.006	0.002

**Table 2 ijms-24-09321-t002:** Comparison of the temperature and pressure of the four-phase equilibrium gaseous carbon dioxide–aqueous solution–ice–carbon dioxide hydrate determined in this study with the literature data [29,32,54].

*T*, K	*P*, MPa	*u_c_*(*T*), K	*u_c_*(*P*), MPa	Determination Method	Reference
271.60	1.044	0.023	0.021	Direct measurement	Mean of runs 1–6 (this work)
271.7	1.03	0.20	0.02	Intersection of V-L_w_-H and V-I-H lines	Yasuda and Ohmura 2008 [29]
271.6	1.04	0.2	0.02	Intersection of V-L_w_(L_w_*)-H and V-I-H lines	Nema et al. 2017 [32]
271.65	1.03	n/a	n/a	Intersection of V-L_w_(L_w_*)-I and V-I(L_w_)-H lines	Mel’nikov et al. 2014 [54]

## Data Availability

Raw data obtained by measuring three-phase V-L_w_-H and four-phase equilibrium coexistence V-L_w_-I-H in H_2_O–CO_2_ system are available at https://data.mendeley.com/datasets/5xjdgvjb84/2 (accessed on 24 January 2023) or on request from the corresponding authors.

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
