# Peer review of "Direct Measurement of the Four-Phase Equilibrium Coexistence Vapor–Aqueous Solution–Ice–Gas Hydrate in Water–Carbon Dioxide System"

_ijms, 2023, doi:10.3390/ijms24119321_

Round 1

Reviewer 1 Report

In the present manuscript, Semenov et al., there is plenty of data available on the hydrate liquid vapour equilibria of hydrate forming systems (in this case H2O-CO2 system) in the literature. However, the data on four phase equilibria (Hydrate-liquid-vapour-Ice) is not sufficiently reported due to some technical issues in its measurements. Therefore, most of the time indirect methods are used and reported. In this work, authors have proposed a direct method to determine the four phase equilibrium of H2O-CO2 hydrate forming systems. This method can also be extended to other hydrate forming systems.

The topic is very relevant to the field and addressed the gap in the literature where a direct method to measure the four phase equilibrium was not available.

The authors have presented a novel direct method to measure the four phase equilibrium of a gas hydrate forming system and compared their results with the one reported earlier by other groups. The method shows improved uncertainty compared to the previously reported methods. The manuscript is flawless in its writing, all the relevant literature is cited properly, experiments were carried out carefully, and the data anlaysis and discussion is thorough. I recommend that the article can be accepted in its current form.

The conclusions in the work are consistent with the evidence and arguments presented in the results and discussion section. Authors did a commendable job by citing appropriate references thoroughly. They paid good attention to the previously published literature. The quality of the figures is good. The information provided in the Tables is sufficient and clear.

Author Response

The authors are grateful to the Reviewer for the detailed analysis of the work and positive evaluation of all components of the article. In our work, we always proceed from the principle that all possible efforts should be made to achieve the best result.

Reviewer 2 Report

The authors have presented new direct method for experimental observation of the quadruple point in the CO2 - H2O system, including vapor, liquid, and two solid phases (Q1). Obtained results for P and T are in excellent agreement with other experimental data taken from literature. Novel data for pressure and temperature at four-phase equilibrium in the binary system under considerartion are useful for those researchers who study P-T diagrams for the mixture in a wide range of conditions involving various phases, thermodynamics of heterogeneous systems, and nonvariant equilibrium. This paper can be published after minor revision noted.

My small comments/remarks are following :

1. Chapter 4. Materials and Methods should be moved to the beginning and placed after Chapter 1. Introduction.

2. Lines 215-216. "the accuracy of the description of the experimental data increases by a factor of 3.5 in the transition from empirical model 1 to 2". It seems to me it is obviously. You can expect more accuracy introducing more fitted/adjustable parameters in eq. 2 in comparison with eq.1. It is better to avoid the phrase "empirical model" here. In fact, we deal with two correlation equations for dependence P(T) describing three-phase line in P-T diagram.

Author Response

1. Chapter 4. Materials and Methods should be moved to the beginning and placed after Chapter 1. Introduction.

Unfortunately, we cannot place Chapter 4. Materials and Methods after Chapter 1. Introduction, since the structure of the paper is determined by the journal template, the use of which is mandatory.

2. Lines 215-216. "the accuracy of the description of the experimental data increases by a factor of 3.5 in the transition from empirical model 1 to 2". It seems to me it is obviously. You can expect more accuracy introducing more fitted/adjustable parameters in eq. 2 in comparison with eq.1. It is better to avoid the phrase "empirical model" here. In fact, we deal with two correlation equations for dependence P(T) describing three-phase line in P-T diagram.

We agree with the Reviewer that increased model accuracy is expected with more fitted/adjustable parameters. We have corrected the sentence on lines 214-217 by adding the phrase "As expected." We have also removed the word "empirical" from the sentences in this paragraph.

Reviewer 3 Report

The document is well-written and presents a good structure. Despite my positive comments, some aspects should be reviewed before publication:

Introduction. 

(i) There are different alternatives to rapidly induce the hydrate formation, such as the incorporation of small amounts (some ppm) of THF, which does not affect the HLV equilibrium of CO2 but reduces the time in the formation process. However, this is not reported in your manuscript. Authors may review: doi: 10.1016/j.cherd.2013.12.007, doi: 10.1016/j.ces.2016.06.034. 

(ii) The authors could indicate that ice formation may benefit the hydrate formation process since this solid phase (ice) is a nucleation center for hydrates.

Results. 

(i) It is suggested to delete Figure 5 because it shows the same information as Figure 4(b). 

(ii) Why is the exothermic picks characteristic of the crystallization of hydrates or ice not seen in Figure 6?

Discussion.

(i) It would be interesting to include potential applications of this technique in hydrate studies.  

(ii) comment on aspects of cost reduction with the methodology proposed in this manuscript.

Materials and Methods: 

(i) The authors should justify why 600 rpm is chosen. Some studies show that high stirring speeds reduce the efficiency of the CO2 solubility process in water.

n/a

Author Response

Introduction.

(i) There are different alternatives to rapidly induce the hydrate formation, such as the incorporation of small amounts (some ppm) of THF, which does not affect the HLV equilibrium of CO2 but reduces the time in the formation process. However, this is not reported in your manuscript. Authors may review: doi: 10.1016/j.cherd.2013.12.007, doi: 10.1016/j.ces.2016.06.034. 

The authors are grateful to the Reviewer for his valuable comment. However, we would like to emphasize that the Introduction section of the paper focuses on different types of phase equilibria in the H2O-CO2 system, i.e., thermodynamics, without detailed consideration of the kinetics of gas hydrates nucleation and growth. Therefore, we did not consider using kinetic promoters to accelerate hydrate formation in the Introduction section. To address this issue, we have expanded the text of Section 3 Discussion.

(ii) The authors could indicate that ice formation may benefit the hydrate formation process since this solid phase (ice) is a nucleation center for hydrates.

The Reviewer raises an issue related to the kinetics of the process. We decided not to mention this issue in the Introduction section for two reasons. First, the Introduction section of the paper focuses on different types of phase equilibria in the H2O-CO2 system, i.e., thermodynamics, without detailed consideration of the kinetics of nucleation and growth of gas hydrates. Second, to the best of our knowledge, the assertion that ice particles can act as nucleation centers of gas hydrates is quite contradictory based on literature data. To confirm or refute this assertion, extensive experimental studies are needed to obtain sufficient statistics.

Results. 

(i) It is suggested to delete Figure 5 because it shows the same information as Figure 4(b). 

We disagree with the Reviewer's assertion that the data in Figure 5 duplicate the information in Figure 4b. The results in Figure 4b show the relative deviations for all experimental points we measured and those reported in the literature. In Figure 5, on the other hand, we can compare how much the average percentage deviation of the equilibrium pressures of carbon dioxide hydrate measured by other authors differs from our values. Therefore, we decided to keep Figure 5.

(ii) Why is the exothermic picks characteristic of the crystallization of hydrates or ice not seen in Figure 6?

Asymmetric peaks caused by the onset of exothermic crystallization of CO2 hydrate are present in the thermal curves of runs 1-5 (Figure 6). A corresponding description is given in the caption of Figure 6. The vertical black dashed lines indicate the onset of CO2 hydrate formation in Figure 6. The nucleation and growth of CO2 hydrate resulted in a temperature jump of 0.13–0.7 K in the autoclave. Ice crystallization resulted in a larger amplitude exothermic effect (0.3–1.8 K temperature jump). The vertical cyan dashed lines indicate the onset of ice crystallization in Figure 6. No temperature change in the form of a peak was observed during ice crystallization because the system quickly reached an equilibrium state (4 phases V-Lw-I-H in equilibrium at T = 271.60 K). The thermal effects of crystallization are discussed in more detail in section 2.2.

Discussion.

(i) It would be interesting to include potential applications of this technique in hydrate studies.  

In the last paragraph of Section 3 Discussion, we have added a few sentences on applying the proposed approach to studying other hydrate-forming systems.

(ii) comment on aspects of cost reduction with the methodology proposed in this manuscript.

Thank you for this suggestion. We have reflected this idea in the text of Section 3 Discussion.

Materials and Methods: 

(i) The authors should justify why 600 rpm is chosen. Some studies show that high stirring speeds reduce the efficiency of the CO2 solubility process in water.

This stirring speed is standard for us when measuring gas hydrate equilibria with the GHA350. We have shown that this stirring rate gives reliable experimental results for a wide range of systems [https://link.springer.com/article/10.1007/s10553-022-01429-w]. We have added a sentence to the Materials and Methods section explaining the selection of the specified stirring speed.

We will be grateful to the Reviewer if he provides us with references showing that higher stirring speeds reduce the efficiency of the CO2 solubility in water. Unfortunately, we are not familiar with these papers.

Round 2

Reviewer 3 Report

Thanks for revising the manuscript based on the suggestions provided in the first round.